# Estimating economic losses to tourism in Africa from the illegal killing of elephants

Robin Naidoo[1,2], Brendan Fisher[3], Andrea Manica[4] & Andrew Balmford[4]

Recent surveys suggest tens of thousands of elephants are being poached annually across Africa, putting the two species at risk across much of their range. Although the financial motivations for ivory poaching are clear, the economic benefits of elephant conservation are poorly understood. We use Bayesian statistical modelling of tourist visits to protected areas, to quantify the lost economic benefits that poached elephants would have delivered to African countries via tourism. Our results show these figures are substantial (~USD $25 million annually), and that the lost benefits exceed the anti-poaching costs necessary to stop elephant declines across the continent's savannah areas, although not currently in the forests of central Africa. Furthermore, elephant conservation in savannah protected areas has net positive economic returns comparable to investments in sectors such as education and infrastructure. Even from a tourism perspective alone, increased elephant conservation is therefore a wise investment by governments in these regions.

[1] World Wildlife Fund US, 1250 24th Street, Washington, District of Columbia 20037, USA. [2] Institute for Resources, Environment and Sustainability, University of British Columbia, AERL Building, Vancouver, British Columbia, Canada V6T 1Z4. [3] Gund Institute for Ecological Economics, University of Vermont, Burlington, Vermont 05405, USA. [4] Department of Zoology, University of Cambridge, Downing Street, Cambridge CB2 3EJ, UK. Correspondence and requests for materials should be addressed to R.N. (email: robin.naidoo@wwfus.org).

The conservation of savannah (*Loxodonta africana*) and forest (*Loxodonta cyclotis*) elephants in Africa is an issue of urgent global significance, as the recent upswing in poaching has resulted in reductions of up to 60% in elephant populations across the continent[1–3]. Demand for ivory, largely to supply Asian markets despite an international commercial trade ban[1], is reducing or eliminating elephants in large swathes of their former range, with recent surveys suggesting tens of thousands of elephants have been poached over the last 5 years from Tanzania and Mozambique alone[4]. Suggested conservation responses to this crisis have included reducing ivory demand in Asia[5,6], increasing incentives for local communities to act as elephant stewards[7] and strengthening the ability of frontline conservationists to prevent elephant poaching[8,9]. The latter two points require range-country governments to amplify their investments in elephant conservation efforts. However, given other pressing development priorities that compete for limited funding and attention, it is typically difficult to justify conservation via a return-on-investment basis, as the tangible economic benefits of biodiversity conservation are rarely understood[10,11].

Here we conduct an economic analysis of the contribution of elephants (grouping both species together) to tourism in Africa's protected areas (PAs). In taking this approach we aim to elucidate how the tourism benefits that are lost due to elephant poaching relate to the enforcement or anti-poaching costs required to prevent elephant population declines that arise from illegal killing. This benefit-cost framework, while addressing an important aspect of elephant conservation and management, is only one small component of what a total economic value study would estimate[12]. In a more comprehensive economic study with greater data availability, additional potential costs such as damages to local communities' crops and the opportunity costs of setting aside PAs[13,14], as well as additional potential benefits such as the ecosystem engineering role of elephants and the existence values that people hold for their conservation[15,16], would all be considered.

Our modelling builds on recent global and continental-scale models of tourist visits to PAs[17] and quantifies the marginal contribution of elephant densities to the expected number of visits to a PA. Conceptually, if fewer elephants are present at PAs due to poaching, and if elephant abundance is indeed an important driver of tourist visits (that is, all else equal, more elephants mean more tourists), the lost economic benefits due to poaching can be estimated as the spending of visitors at and near PAs that will no longer occur due to reduced visitation rates. To make such a valuation, we use information on the average number of annual visits to 164 PAs within 25 elephant range-state countries (these 25 countries collectively contain > 90% of Africa's elephants), including 110 PAs that contain elephants

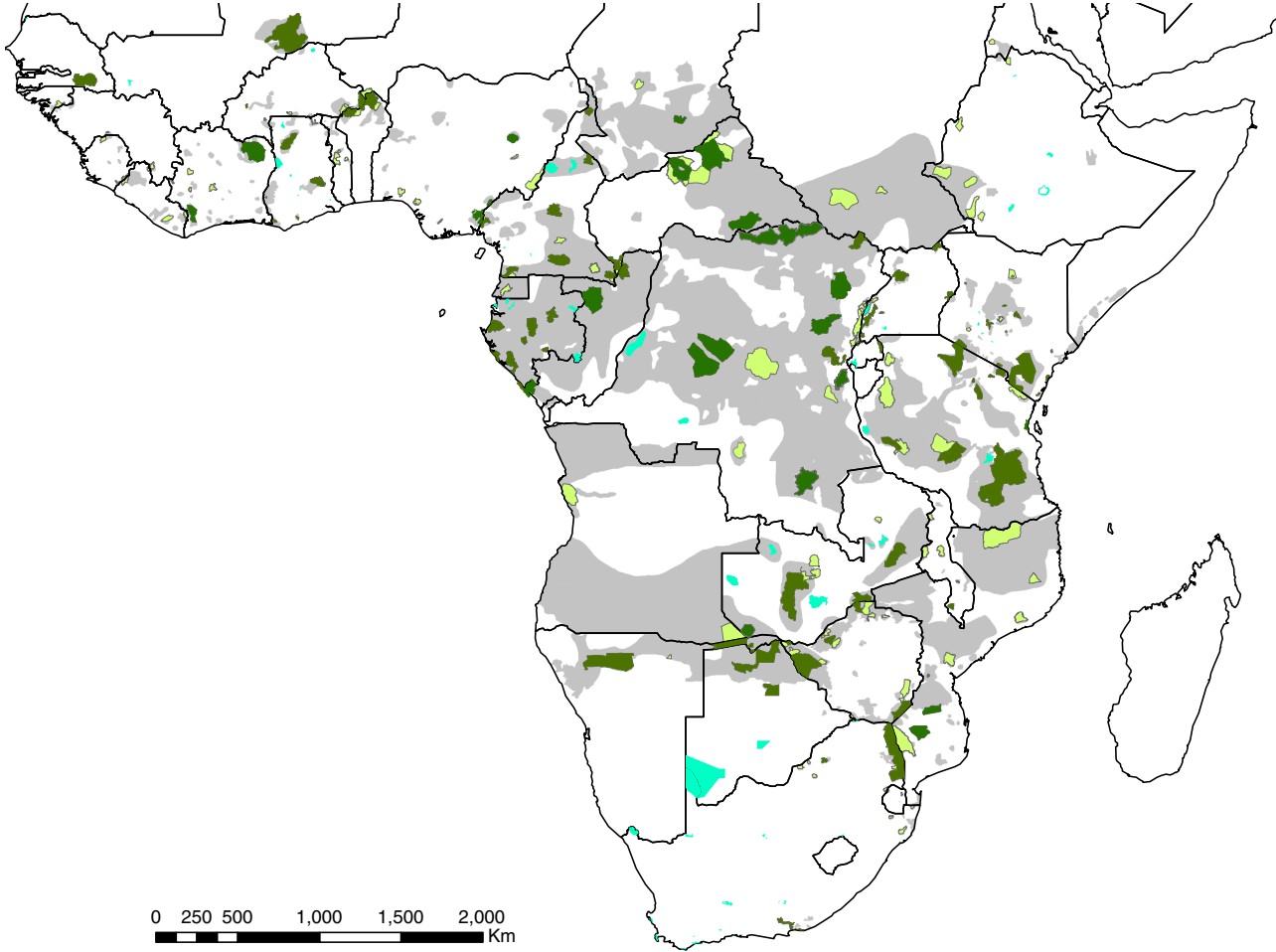

**Figure 1 | Protected areas and elephant distribution in Africa.** Combined range of the two African elephant species (grey), with International Union for Conservation of Nature Category II–VI protected areas[46] that harbour elephants (dark green indicating *n* = 110 that have tourist visitor data and light green indicating those without visitor data) and protected areas with no elephants for which we have tourist data (blue, *n* = 54).

(Fig. 1). In addition, we harnessed information on the most recent (typically *ca.* 2009–2013) comprehensive population estimates (http://elephantdatabase.org)[18] and rates of illegal killing at 216 PAs, and on the average direct and indirect spending levels of nature-based tourists visiting PAs in Africa (see Methods). Our figures for the per-unit-area spending necessary to effectively curtail elephant poaching are derived from empirical modelling work conducted at the height of the first wave of African elephant poaching during the 1980s (refs 19,20); to our knowledge, there have been no similarly thorough estimates derived during the current poaching crisis.

We find that the lost economic benefits that elephants could deliver to African countries via tourism are substantial (∼USD $25 million annually), and that these benefits exceed the costs necessary to halt elephant declines in east, southern and west Africa. Even if we entirely ignore other benefits that people derive from elephants[21], their conservation is a wise investment decision for countries in the savannah regions of Africa, although not currently so in the forested regions of central Africa.

## Results

**Aggregate impact and valuation of elephant losses to tourism.** The tourism model we developed explains 44% of the variance in visitation rates to Africa's PAs (Fig. 2). After controlling for a number of other potentially confounding variables, there was very strong support (95% Bayesian credible intervals that do not overlap with zero) for elephant density as a positive predictor of the annual number of visits a PA receives (Table 1). There was also very strong evidence of an interaction between elephant density and whether PAs were forested or savannah (the positive effect of elephants on visits was much reduced in forests), a negative effect of PA size and a positive one of country-level wealth. In addition, there was substantial support (90% Bayesian credible intervals that do not overlap with zero) for the impacts of surrounding population (negative; PAs with smaller surrounding populations had more visits), the presence of another charismatic megafauna species, the lion *Panthera leo* (positive; PAs with lions had more visits) and a main effect of forested PAs

(negative; fewer visits to forest PAs than savannah PAs). After controlling for all these independent variables, our model showed that a 1-unit increase in elephant density resulted in a $100 \times (e^{1.55} - 1) = 371\%$ increase in PA tourist visits. At the median number of PA visits in our data set (1,883), this result implies that an increase in elephant density of $0.1\,\mathrm{km}^{-2}$ resulted in an additional ∼700 annual visits to a PA, all else equal.

We used our model to predict tourist visitation rates at all 216 PAs in Africa that currently harbour elephants. We then used population-specific estimates of changes in elephant densities[1], to estimate the annual number of elephants being lost to poaching at each PA, and simulated how this loss would reduce annual tourist visits by re-running our model using these new predicted elephant densities. To monetize the reduction in the flow of tourists to PAs due to elephant poaching, we simulated economic losses resulting from direct spending (using a best-fitting exponential distribution parameterized from 36 estimates of in-country, per-visit expenditure on nature-based tourism in Africa; Supplementary Fig. 1) and also from indirect and induced spending (using a best-fitting Gaussian distribution parameterized from 24 studies that estimated local economy 'multiplier' impacts of African nature-based tourism; Supplementary Fig. 2). We drew independently from each of these distributions for each PA, multiplied these values by the estimate of annual losses in tourist visits and repeated 100,000 times.

Using this valuation procedure we estimate that across Africa the annual, direct economic losses from reduced PA visitation due to elephant poaching run to a mean of $9.1 million (USD 2016; 95% Bayesian credible interval (CI) $4.86–$15.7 million), with an additional mean loss of $16.4 million (95% CI $8.56–$28.9 million) in indirect and induced spending. These estimates represent the first continent-wide assessment of the economic losses that the current elephant poaching surge is inflicting on nature-based tourism economies in Africa. Using a central figure of ∼$25 million in lost economic benefits per year highlights the relative impact of these losses: this represents close to 20% of the receipts from all PA visits in 14 countries that contain half of Africa's elephants[22] and, tabulating ecoregion-level costs of effective biodiversity conservation[23], ∼7% of the funding required to conserve biodiversity in ecoregions in which elephants occur. On the other hand, the economic difficulties of elephant conservation are also illustrated by the fact that annual losses to tourism are only a small fraction of the estimated $597 million that ivory from Africa's poached elephants was worth annually on Chinese black markets from 2010–2012 (see Supplementary Note 1).

**Geographic variation in tourism loss from elephant poaching.** Disaggregating the overall figures for the economic losses associated with poaching of elephants at PAs across the continent reveals substantial variability in their geographic distribution. Regionally, the greatest losses occur in east and southern Africa (Table 2). This is driven not by poaching rates, which are actually substantially lower in those regions than in central Africa[1–3], but rather by high visitation rates to PAs and the fact that the positive impact of elephant density on tourism visits is strongly reduced in the forested PAs of central Africa (Table 1). As such, the aggregate current tourism expenditures that are lost due to elephant poaching in central African forested PAs are negligible ($0.009 million with 95% CI $0.02–$0.05 million), but are several orders of magnitude higher in east Africa (mean $12.2 million; 95% CI $4.17–$27.8 million) and in southern Africa (mean $13.0 million; 95% CI $5.69–$24.8 million). These lost tourism benefits due to elephant poaching can be a substantial fraction of all

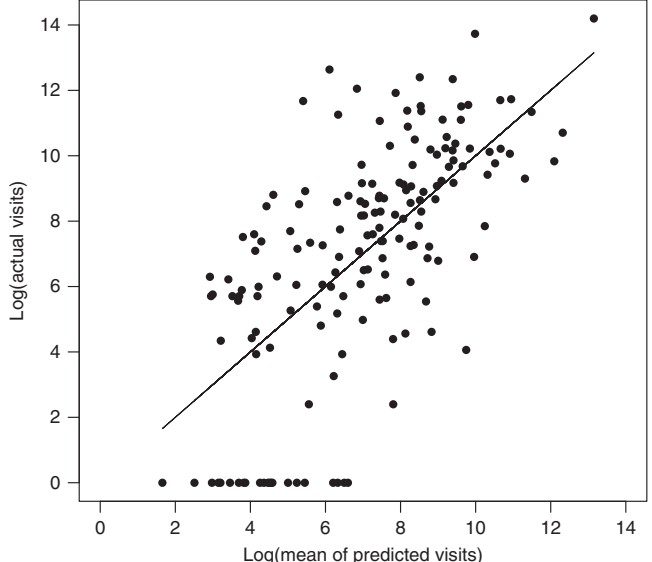

**Figure 2 | Evaluating the predictive value of the tourist visitation model.** Actual (*y* axis) versus median predicted (*x* axis) average annual tourist visits (log-transformed) from a Bayesian regression model of tourist visits to 164 protected areas in Africa. Regression equation: $Y = 1.02 \times X - 0.14$, $R^2 = 0.44$, $P < 0.0001$.

nature-based tourism in countries where savannah, rather than forested, ecosystems predominate. For example, in Tanzania, we estimate that the average total tourism benefits lost due to elephant poaching are ~$540,000 per year, or between 4% and 11% of the total receipts from all visitors to PAs (estimated at $5–$15 million per year[22]).

**Return-on-investment from elephant conservation.** How do the lost benefits from reductions in elephant-based tourism compare with the costs that would be required to reduce or eliminate the poaching of elephants (and therefore sustain these benefits) at PAs across the continent? Few studies have analysed anti-poaching costs, in particular with varying effectiveness targets and across large scales encompassing different habitat types. The only such study we are aware of was conducted during the height of elephant poaching in the 1980s and developed a regression model of the relationship between changes in large ($>1,000$) elephant populations and per-unit-area investment in conservation across 14 African countries[19,20]. To achieve no decline in elephant populations required spending levels of $215 \, \text{km}^{-2}$ in USD 1981, equivalent to $565 \, \text{km}^{-2}$ in USD 2016 (converted using the United States' Department of Labor Consumer Price Index inflation calculator; http://data.bls.gov/cgi-bin/cpicalc.pl). We used this regression model to estimate the shortfall (based on changes in PA-specific elephant populations from illegal killing) required to stabilize elephant populations for each of 58 PAs containing over 1,000 elephants.

The overall costs for reducing poaching to a level that rendered elephant populations stable (that is, no growth but no decline) in PAs with large elephant populations were estimated at $26.5 million annually across the 58 PAs, with almost two-thirds of this cost ($16.9 million) occurring in the large, mostly forested PAs of central Africa where poaching has been heaviest (Table 2).

Comparing these costs with the total lost tourism benefits due to elephant poaching at the same sites reveals average rates of return (the difference between average benefits and costs, divided by the costs) on elephant conservation that are highly negative in central Africa ($-100\%$, because of a large shortfall in spending and few visitors), positive in west Africa (16%; modest visitation but also—because of low elephant numbers—a limited spending gap), and strongly positive in southern Africa (54%) and east Africa (78%; where gains in visitor spend would substantially outweigh the necessary increases in anti-poaching expenditure). From a regional, return-on-investment point of view, elephant conservation in the savannah PAs of east, southern and west Africa is justifiable based on the economic returns from tourism alone. The average rate of return on elephant conservation in these regions also compares favourably with estimated rates of return to investments in education[24], agriculture[25], electricity[26] and infrastructure[26] that governments in African elephant range countries routinely make (Fig. 3).

**Changes in elephant density and tourist visits over time.** Our results are based on across-site variation in tourism visits and changes in elephant densities for a large set of African PAs. How do these results compare with changes in visits and elephant numbers within a single site? Acquiring a large panel data set on changes over time in tourists, elephant densities and additional covariates across many PAs would have been ideal, but in practice we were only able to locate one PA with sufficient data to do a within-site comparison. Addo Elephant National Park in South Africa has a published time series from 1954 to 2010 on elephant numbers and visitors[27], and a bivariate plot of the two indicates a general positive relationship split into two distinct phases (1956–1995 and 1996–2010; Supplementary Fig. 3). Although our model of tourist visits across African PAs mostly contains

---

**Table 1 | Bayesian regression model results.**

| | Mean | s.d. | 2.5% | 97.5% | Number effective samples | R-hat |
|---|---|---|---|---|---|---|
| Intercept | 6.38 | 4.18 | −1.83 | 14.49 | 4479 | 1 |
| Area | −0.83 | 0.36 | −1.54 | −0.11 | 21508 | 1 |
| Elephant density | 1.55 | 0.39 | 0.80 | 2.32 | 3633 | 1 |
| Forest | −1.26 | 0.68 | −2.59 | 0.05 | 3181 | 1 |
| Elephant density × forest | −2.02 | 0.69 | −3.39 | −0.73 | 2098 | 1 |
| Lion | 1.00 | 0.58 | −0.14 | 2.14 | 27139 | 1 |
| Natural attractiveness | −0.18 | 0.34 | −0.84 | 0.49 | 18595 | 1 |
| Nearby human population | −0.34 | 0.21 | −0.75 | 0.07 | 18652 | 1 |
| Accessibility | −0.70 | 0.49 | −1.65 | 0.25 | 3595 | 1 |
| Country PPP | 2.05 | 0.53 | 1.02 | 3.08 | 4588 | 1 |

Bayesian regression model results for a model of the average number of annual tourist visits (log-transformed) across 164 protected areas in sub-Saharan Africa. The mean, s.d., 2.5% quantile and 97.5% quantile of posterior coefficient estimates are presented, as well as the number of effective samples and the R-hat measure of parameter convergence. PPP, Purchasing Power Parity.

---

**Table 2 | Estimating the lost tourism benefits from the illegal killing of elephants.**

| Region | All PAs with elephants | | | | PAs >1,000 elephants | | | | |
|---|---|---|---|---|---|---|---|---|---|
| | Predicted annual visits | Direct tourism benefits lost* | Indirect/induced tourism benefits lost* | Total benefits lost* | Cost to maintain population* | Direct tourism benefits lost* | Induced tourism benefits lost* | Total benefits lost* | Rate of return (%) |
| Central | 8,412 | 0.003 | 0.006 | 0.009 | 16.9 | 0.003 | 0.005 | 0.008 | −100 |
| East | 384,439 | 4.37 | 7.83 | 12.2 | 3.29 | 2.13 | 3.83 | 5.31 | 78 |
| South | 1,605,487 | 4.64 | 8.32 | 13.0 | 6.14 | 3.45 | 6.19 | 5.52 | 54 |
| West | 55,405 | 0.12 | 0.22 | 0.34 | 0.14 | 0.06 | 0.10 | 0.16 | 16 |

PA, protected area.
Regional distribution of predicted annual visits and lost tourism benefits across all PAs containing elephants, as well as investment costs required to halt poaching, lost tourism benefits and rates of return across 58 African protected areas that contain large (>1,000) elephant populations.
*2016 USD millions.

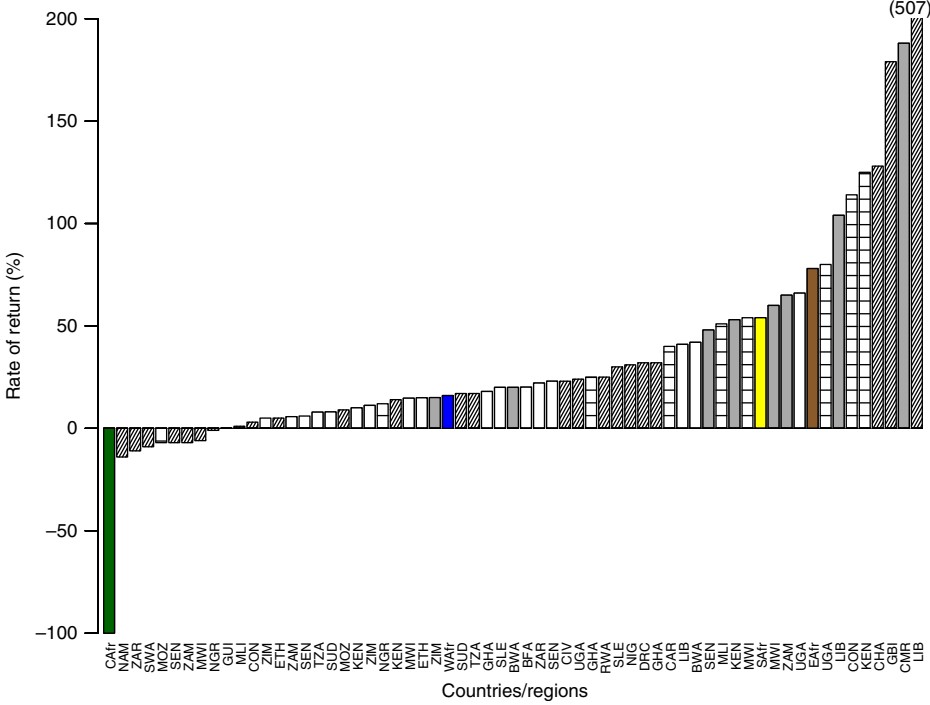

**Figure 3 | Comparing rates of return in African elephant conservation to other investments.** Mean rates of return to tourism from investing in efforts to reduce elephant poaching in central (green), west (blue), southern (yellow) and east (brown) Africa, along with a sample of rates of return estimated for investments in education (white bars[24]), agriculture (grey bars[25]), electricity (horizontal cross-hatched bars[26]) and infrastructure (angled cross-hatched bars[26]) in 33 African elephant range state countries (country abbreviations on bottom axis of figure).

**Table 3 | Models of tourist visitation and elephant densities at Addo Elephant National Park.**

|  | Mean | s.d. | 2.5% | 97.5% | Number effective samples | R-hat |
|---|---|---|---|---|---|---|
| *1965–2010* | | | | | | |
| Intercept | 10.91 | 7.09 | − 2.94 | 24.91 | 1,490 | 1 |
| Area | 0.47 | 0.1 | 0.27 | 0.66 | 2,104 | 1 |
| Country GDP | − 0.46 | 0.9 | − 2.22 | 1.3 | 1,494 | 1 |
| Elephant density | 0.67 | 0.21 | 0.26 | 1.08 | 1,355 | 1 |
| $\sigma$ (tourists) | 0.38 | 0.04 | 0.31 | 0.47 | 2,456 | 1 |
| *1965–1995* | | | | | | |
| Intercept | 13.12 | 6.38 | 0.72 | 25.2 | 1,661 | 1 |
| Area | 0.22 | 0.09 | 0.05 | 0.4 | 1,978 | 1 |
| Country GDP | − 0.59 | 0.81 | − 2.13 | 0.99 | 1,652 | 1 |
| Elephant density | 0.55 | 0.18 | 0.19 | 0.9 | 1,549 | 1 |
| $\sigma$ (tourists) | 0.3 | 0.04 | 0.23 | 0.39 | 1,864 | 1 |
| *1996–2010* | | | | | | |
| Intercept | 9.72 | 0.58 | 8.56 | 10.87 | 2,011 | 1 |
| Elephant density | 0.66 | 0.19 | 0.27 | 1.05 | 1,978 | 1 |
| $\sigma$ (tourists) | 0.15 | 0.04 | 0.09 | 0.26 | 1,560 | 1 |

Bayesian regression model results for a model that predicts annual tourist visits (log-transformed) at Addo Elephant National Park in South Africa. The mean, s.d., 2.5% quantile and 97.5% quantile of posterior coefficient estimates are presented, as well as the number of effective samples and the R-hat measure of parameter convergence.

variables that are time invariant (that is, forest/non-forest, year of establishment, access, surrounding population and natural attractiveness), country gross domestic product (GDP) and park area did change over time at Addo. We therefore built a subset of our main model by regressing tourist visits against park area, country GDP and elephant density. Despite this different set of independent variables, the coefficient on elephant density remained positive, with Bayesian credible intervals above zero and overlapping the range of our across-site analysis (0.67, with 95% CI of 0.26–1.08; Table 3). The coefficient on elephant density

remained positive when we restricted the analysis to the 1956–1995 period (0.55; 95% CI 0.15–0.9) and also for the 1996–2010 period (0.66; 95% CI 0.27–1.05; it is noteworthy that here we regressed tourist visits on elephant density alone, as the sample size was too small to include the other independent variables). These results suggest a degree of concordance between the among- and within-site analyses (Supplementary Note 2), although Addo may only be representative of those PAs that, similar to itself, are fenced and where elephant populations have been strictly managed. Additional data at non-fenced, less heavily

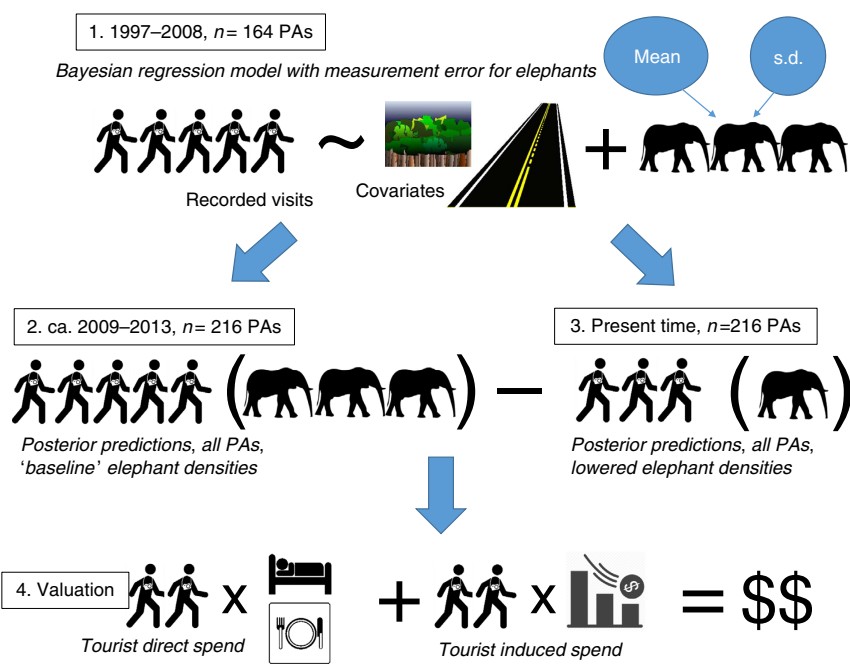

**Figure 4 | Workflow representation of methods.** Our modelling methodology involved the following steps: (1) Bayesian model estimation of tourist visits at 164 PAs; (2) predicted visits at 216 PAs, no elephant poaching; (3) predicted visits at 216 PAs, with reduced elephant densities from poaching; (4) stochastic valuation of lost visits via distributions of the direct and indirect spending of tourists at PAs.

managed PAs would have been useful to further assess the generality of our results for individual sites.

## Discussion

Although our results make use of comprehensive, spatially explicit data on elephant densities at PAs across Africa, emerging results from the most recent census efforts suggest that declines in elephant populations in some countries have been even steeper than those previously documented[1,28]. As such, our estimates of the economic losses to tourism from elephant poaching may well be conservative, although we also understand that tourism takes time to evolve in places, and that responses to elephant decreases will not happen immediately at any given site. Despite the recognized importance of Africa's natural assets, especially wildlife, to tourism and other development pathways[29], our analyses were limited by the amount, quality and spatial resolution of data on the nature-based tourism sector. This was particularly true for expenditures that tourists make during PA visits and the associated impacts this injection of money can have in local economies[30]. Finally, the increasing magnitude and sophistication of elephant poaching may necessitate higher per-unit-area anti-poaching costs in heavily hit areas, although the deployment of novel, high-tech solutions such as unmanned aerial vehicles and infrared remote cameras may simultaneously drive costs down[31]. Anti-poaching costs also no doubt vary across sites due to other ecological and socioeconomic factors, but data to address this variation are sorely lacking and we were therefore obliged to rely on rigorous but dated information on anti-poaching costs collected during the 1980s wave of elephant poaching in Africa. Moreover, although anti-poaching efforts have strong positive impacts on elephant populations in both forest[32] and savannah[8] systems despite being generally underfunded across African PAs[33,34], they are not the only site-level actions that are important for elephant conservation[35].

Despite these caveats, our results suggest two broad conclusions. The first is that elephant conservation in PAs of the savannahs of Africa represents a wise investment with immediate and ongoing payback for tourism. Rates of return are positive, sometimes strongly, in these areas, indicating that tourists' willingness to pay, to see elephants as part of a visit to a PA, are sufficient to offset the increased costs necessary to safeguard elephant populations. These results align with surveys that have shown that elephants are among the most desired of African wildlife species for tourist viewing[36,37], suggesting that declines in elephants from poaching drive tourism losses, rather than the converse. Anecdotal information on the impacts of the even more catastrophic recent losses of elephants across Africa also suggests that tourism is under threat or has already declined (see Supplementary Note 3).

The second conclusion is that elephant-based tourism cannot currently be expected to contribute substantially to the conservation of forest elephants in central Africa. In these remote, difficult-to-access areas where tourism levels are currently lower than in savannahs and where elephants, with few exceptions[38], are difficult to see, different funding mechanisms that capture public concern and the 'existence value' of elephants will be necessary to halt recent declines[2]; examples include the Partnership to Save Africa's Elephants (a Clinton Global Initiative) and the Elephant Crisis Fund. Global forest-based conservation schemes, such as Reducing Emissions from Deforestation and Degradation (REDD+), may also have a role to play if associated biodiversity considerations, such as the conservation of elephants, can be incorporated[39]. Our results additionally highlight that the conservation of biodiversity cannot always be justified from a purely financial point of view, and that the 'use values' or 'ecosystem services' that biodiversity provides are complementary to, rather than substitutes for, moral or aesthetic reasons for conservation[40].

Although the value of ivory from poached elephants on Chinese black markets swamps that of the resulting losses in tourism, ivory benefits are not realized by governments or the people of African range states, apart from the few that are involved in the illegal killing. In contrast, tourism benefits from

elephant conservation have the potential to reach a much broader cross-section of Africans, although financial considerations, such as the profit margins of tourism operators and the ability of policy makers to channel revenues from tourism to key stakeholder groups, are obviously critical to ensuring these net benefits are translated into effective conservation action. In particular, it will be fundamental to ensure that local communities and landholders are sufficiently incentivized to embrace living alongside elephants, or at minimum, are sufficiently compensated so as to not collaborate with poaching syndicates (for example, see www.ecoexistproject.org)[41]. Although there is a long history of nature-based tourism benefits not reaching local communities[42], recent experiences in African elephant range countries have demonstrated some successes in the devolution and capture of benefits from local natural resource management[43,44]. Ensuring that those who live with elephants are sufficiently compensated and motivated to do so, whether via tourism or other avenues, will play a central role in the success or failure of Africa's elephant conservation efforts.

## Methods

**Tourism data.** For tourist visits to African PAs, we extended the visitor database of a recent global study[17], compiling data at additional PAs from published research, the grey literature and personal contacts familiar with tourism in various regions across sub-Saharan Africa. This resulted in a database with information on annual visitation rates for 164 PAs that occur in countries that contain African elephants. We simultaneously searched the literature for estimates of the economic importance of tourism visits to PAs in Africa via (1) the direct, in-country expenditure (not including the costs of international airfare) that a tourist spends at a PA[17,22] and (2) the economic impact, or 'multiplier' effect, that a tourist dollar has as it trickles through the local economy after its initial expenditure[30,45]. We were able to compile $N = 36$ and $N = 24$ such estimates, respectively, which we used in valuation simulations as described in the main text and below (Supplementary Figs 1 and 2).

**Elephant and PA data.** We extracted data on the size and location of elephant populations across Africa from the African Elephant Database (http://elephantdatabase.org)[18]. G. Wittemyer kindly provided annual growth rates (ca. 2012), including the proportion of elephants killed illegally (PIKE), for these same populations[1]. We cross-referenced these spatial estimates of elephant populations and their growth rates with International Union for Conservation of Nature (IUCN) PAs in categories II–VI (excluding category I PAs where tourism is largely prohibited) using the World Database on Protected Areas[46], extracting all PAs that overlap with known elephant populations (Fig. 1). For each of these 216 PAs, we extracted information on additional potential predictors of tourist visitation rates as per the model in ref. 17 and as described below.

**Modelling PA visitation rates.** We built a model of the average annual number of visits to PAs across the African elephant's range using 164 PAs for which we had information on tourist visits, elephant populations and a set of additional predictor variables previously used in modelling tourist visits to PAs[17]. Briefly, these additional variables were as follows: (1) PA size—we expected larger PAs to have more visitors; (2) surrounding population—we expected PAs with more people living around them to have higher numbers of visitors; (3) accessibility—we expected more accessible PAs (measured by the minutes to get to the PA over land and/or water routes from the nearest large city) to be more heavily visited; (4) national income (2006 PPP)—we expected richer countries to have greater levels of PA visits; and (5) natural attractiveness—we expected PAs with a higher such score (measured subjectively as a 1–5 index of the attractiveness of the birds and mammals a visitor might expect to observe for 65 biome-realm combinations) to have more visitors. In addition to the predictors in ref. 17 and elephant density (the result of the stochastic draw of elephant population mean divided by the area that was censused at each PA), we also included the interaction between elephant density and forest/non-forest land cover type (based on an assessment of the dominant land cover contained within each PA[47]), as we expected elephants to be less important draws for tourists in forested areas where they are difficult to observe. In addition, and recognizing that other charismatic megafauna have the potential to drive tourism, we used recent and comprehensive rangewide distributions[48,49] of the lion *P. leo* to include lion presence/absence at a PA as a further predictor in our visitation models. Our previous work[17] investigated other possible variables of importance that were ultimately not included in the final visitation model (for example, distance to major airport and incidence of armed conflict) and data availability constraints precluded other potential drivers, such as the activities on offer at a park (for example, mountain biking, hiking and fishing), from being included.

As our tourism data were almost entirely from 1998–2007 (ref. 17), we used elephant population estimates that overlapped with this time period where possible (75% of cases). We used a Bayesian regression modelling approach that offered several advantages to traditional/frequentist multiple linear regression methods. First, elephant populations are estimated with uncertainty and our Bayesian framework explicitly incorporated this uncertainty by using as the predictor a normal distribution for the population at each PA (defined by the mean and s.d.), rather than a point estimate as required by ordinary multiple linear regression[50]. A majority (55%) of the elephant populations had estimates of the uncertainty around the population size, expressed either as a s.d. from an assumed normal distribution (43%) or as a range (12%). In the latter case, we assumed a normal distribution centred around a mean at the midpoint of the range and assumed the range endpoints represented 95% CIs, following best practice in such instances[50]. A second advantage of Bayesian regression methods is that for the 45% of population estimates where no uncertainty estimate was provided, these missing values can be stochastically and simultaneously imputed within the same model, using the strong positive linear relationship we observed between the s.d. and mean of elephant population sizes (s.d. $= 192 + 0.122 \times$ mean, $n = 49$, $R^2 = 0.85$).

We (natural) log-transformed tourism visits so that the resulting distribution better approximated the normal and also log-transformed most of the predictor variables (Supplementary Table 1) to reduce differences in scale that could affect the Bayesian estimation procedures[51]. We used the modelling language Stan and the R statistical computing software to develop our models, using 4 Monte Carlo chains of 25,000 iterations after a 25,000-iteration warmup period each, for a total of 100,000 samples (a figure necessary to stabilize resulting value estimations that we derived from preliminary trials). Priors on all estimated parameters were uninformative[51]. We assessed convergence of the chains by ensuring that effective sample sizes were large and by ensuring that the potential scale reduction statistic, R-hat, was ≤1.01 for all estimated parameters[52]. See Supplementary Note 4 for a more detailed exposition of the model. We did not define a threshold cutoff for statistical 'significance', but rather interpret variable coefficients where 95% Bayesian credible intervals do not overlap with zero as providing very strong evidence for a variable's impact, with more moderate support for variables whose 90% Bayesian credible intervals do not overlap with zero.

**Economic valuation of elephant losses from poaching.** We used our Bayesian regression model to generate posterior predictions on the impact of the most recent reductions in elephants due to illegal killing. We first updated our estimates of tourist visits at all 216 PAs that contain elephants by holding all variables at their mean values and generating a set of predictions for tourist visits reflecting elephant population numbers from the most recent round of elephant censusing at sites across Africa (typically ca. 2009–2013, as opposed to the ca. 1998–2007 population estimates that coincided with our tourism data, and that we used to parameterize the PA visits model). We then used the site-specific PIKE estimates to calculate current annual reductions in elephant densities due to poaching for each site and, keeping all other predictor variables at their mean values, used these new, lowered elephant densities to generate predictions of annual visitation rates to all 216 PAs under expected current changes in elephant densities due to poaching. Taking the difference between the median visits with and without PIKE at each PA and summing these estimates resulted in a range-wide, aggregate annual reduction in PA visitation rates due to current illegal killing rates of elephants of ∼12,500 tourist visits.

As described in the Results, we then monetized this reduced flow of tourists by drawing direct expenditure values and indirect/induced multiplier effects from their respective distributions for each PA, multiplying these values by the predicted reduction in the PA's tourist visits due to elephant poaching and repeating this 100,000 times. Figure 4 provides a pictorial summary of the methods we used to assess the impact of elephant poaching on PA visits and the subsequent valuation of these visit losses.

To estimate the investment necessary to prevent the illegal killings of elephants calculated to be occurring at each PA, we drew on the only published studies that have quantitatively assessed the relationship between per-unit-area anti-poaching costs and changes in elephant populations across multiple study sites and habitat types[19,20]. These studies were conducted across 14 African countries (covering both forest and savannah habitats) during the height of the first wave of elephant poaching in the 1980s and resulted in a regression model of change in elephant population size as a function of per km² conservation expenditure. The model is applicable to large elephant populations ($>1,000$ individuals) and resulted in an estimate of \$215 km$^{-2}$ in 1981 USD (\$565 km$^{-2}$ in 2016 USD) in conservation spending necessary to prevent elephant declines. We view this \$565 km$^{-2}$ estimate as conservatively high, given that it is several times higher than site-level cost estimates to halt elephant poaching in Zambia[53] and in Ghana[54].

Using this benchmark cost estimate of \$565 km$^{-2}$, we estimated the shortfall in spending that would be necessary to reduce the illegal killing of elephants at PAs containing $>1,000$ elephants such that populations were in equilibrium, by: (1) using PA-specific estimates of changes in elephant populations under current PIKE levels to generate, via the regression equations in refs 19,20, the expected amount of km$^{-2}$ conservation spending occurring at each site[1] and (2) subtracting these spending estimates from the \$565 km$^{-2}$ benchmark level. This resulted in conservation spending estimates that would be required to stabilize elephant populations for the 58 PAs that contained $>1,000$ elephants, which we then

compared against the direct and total tourism benefits lost due to elephant poaching at these same sites.

**Data availability.** The data used to build the Bayesian regression models of tourist visitation to African PAs are given in Supplementary Data 1. Additional data are available from the authors upon request.

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

## Acknowledgements

We thank George Wittemyer for providing population-specific data on elephant lambda rates and for helpful comments on the study design. Lisa Steel, Allard Blom, Bas Huijebregts, Paya deMarcken, Mesmin Tchindjang, Shelley Preece, Erica Rieder, Flip Nel, Peter Lindsey and Martha Bechem provided data on tourism visitation rates. We thank Colby Loucks, Jim Sano, Jeff Parrish, Chris Thouless, Fiona Maisels, Hugo Jachmann, Lamine Sebogo, Julie Thomson, Louise Gallagher, George Powell and Chris Weaver for advice on the study and/or comments on earlier versions of the manuscript.

## Author contributions

R.N. conceived of extending the original protected area visitation model of A.M. and A.B. R.N., A.M. and A.B. compiled the data. R.N. analysed the data with input from B.F., A.M. and A.B. R.N., B.F., A.M. and A.B. wrote the paper.
