## [Peer Review File · Nature Communications]

Reviewers' comments:

Reviewer #1 (Remarks to the Author):

In summary: interesting study, worthwhile approach, definitely publishable, but as currently written, has some significant shortcomings which would need to be addressed through a fairly substantial revision.

My interests: familiar with elephant conservation, poaching, tourism, Africa and Asia. Not expert in Bayesian statistical modelling.

Concerns: (a) context, and (b) analysis.

Context.

The ms is framed as an economic calculation at regional scale, but its conclusions suffer from inadequate consideration of regional context and history. It is calculated using a private-actor micro-economic approach: short-term cash spent on anti-poaching, short-term cash received through tourism.

So essentially it concludes that if (and only if) there are lots of wealthy tourists, relatively few elephants, and tourists differentially choose destinations with more elephants, then anti-poaching investments are profitable.

That analysis, however, only applies at local scale. In fact, even where private conservation reserves do indeed have to make exactly that calculation for their anti-poaching investments, they take a much longer investment timeframe.

At national scale, any form of tourism takes time to grow, to establish reputation and distribution networks and infrastructure. There is a wildlife tourism industry in East and South Africa because of anti-poaching investments in the past, and there will only be future wildlife tourism sectors in Central and West Africa if anti-poaching investments are made now.

Of course, the authors of this ms are very familiar with these issues, but the way the ms is presented, these key contextual issues are barely considered.

Analysis.

Neither the ms nor the Supplementary Information present the actual raw data. I think these should be provided. I also think that a single analysis including all parameters simultaneously runs substantial risk of analytical errors.

I would like to see the actual data, tabulated park by park, and including any secondary or estimated parameters calculated or interpolated from subsidiary models as in the SI.

Then I would like to see standard stepwise multiple linear regressions for each of the four regions, so we can judge the relative effects of different factors in different regions. At present these regional differences are hidden in Table 2, which indicates that "cost to maintain population" is 120 times higher in Central than West Africa, which seems improbable.

Currently, the conclusions for South and West Africa are surprising and very counter-intuitive. Therefore, they need to be supported very clearly and carefully by data and calculation which readers can follow, test and replicate.

Yes, linear regression models are unfashionable and Bayesian models are fashionable, but the difference between the underlying model philosophies are likely to have far less effect on the outputs than factors such as data deficiencies and transformations.

For example, why is "year established" log-transformed (ST1)? Why not "years since establishment"? And how can the log of a year be 7.6? That means that the year is $10^{(exp) 7.6}$?!!!

How is "minutes to nearest city" measured? By road? Or by light aircraft, which is how the higher-spending tourists arrive? And why this particular measure, when data from tourism literature indicates that tourists will visit anywhere within 2 hours of a major airport, and visitation falls off heavily beyond that?

How is attractiveness measured on a scale from 2.83 to 5.34? After safety and security, which are not considered at all, attractiveness is the dominant driver of tourist destination choice. But here it is treated equally with Country PPP, USD, log-transformed.

Why is elephant density NOT log-transformed, when the distribution appears to be highly skewed (minimum 0, mean 0.48, maximum 5.10)?

Specifics.

Line 121. Economic losses driven by visitation rates not poaching rates. Standard problem with

snapshot microeconomics. Eg, travel-cost valuations of parks, wilderness, forests etc appear higher for those where off-road vehicles, snowmobiles etc are permitted, because those are expensive, even though they create damage so conservation values are lower.

Line 153. "Savanna PA's of West Africa"? Which countries, which PA's? Very different from those of East and South.

Line 210. What about African Parks?

Line 240. What about EcoExist?

Fig 2. Why has the regression line been forced through (0,0)? Why are the 8 points with log (actual) = 0 included in the regression? Is the y-axis really log [(actual visits)], or log [(actual visits) + 1]? If the latter (which I suspect, since records are hardly likely to include a single visitor per annum), then the regression line should not be forced via (0,0), and the 8 zero-visitor parks should be excluded, which would yield a very different regression line, changing the entire analysis.

Fig 3. Why have industry sectors other than tourism been shown by individual country, but tourism only by region? Why are the RoI shown in increasing succession, rather than by sector or region?

Reviewer #2 (Remarks to the Author):

Review of Naidoo et al. 'Estimating economic losses to tourism in Africa from the illegal killing of elephants'.

GENERAL COMMENTS

I enjoyed reading this study that used Bayesian statistical modelling of tourist visits to protected areas to demonstrate how losing elephants from poaching would substantially decrease the benefits of elephant conservation to tourism in Africa's protected areas. The authors also assessed how these benefits compared to the costs of counteracting poaching. The authors have a strong publication record both in assessing the economic benefits derived from biodiversity and ecosystem services and in assessing factors affecting visitation rates in protected areas both in Africa and globally. The authors have used similar statistical analyses and modelling techniques in previous papers and this is reflected in the fact that the Methods (for an expert) are sound and difficult

to argue against. However, I have a number of concerns that need to be addressed in a revised manuscript.

1) The biggest assumption in the study is that elephant are the only attractor of tourists to Africa's protected areas. As the authors know very well, elephant is indeed one of the charismatic species tourists are after, but not the only one. Lion is at least as important as elephant for ecotourism and similarly declining throughout its range. So are other charismatic species (eg leopard). Hence, I wonder (i) whether your regression model should have included other charismatic species' density as a predictor variable and (ii) whether the estimate of loss is realistic considering you didn't include other charismatic species in the analysis? I think this is an important limitation of the study and needs to be explained better. I understand data are available mainly for elephant, partly because of the recent study by Wittemyer and colleagues that you cite, but I think, considering the literature on tourists' preferences in sub-Saharan Africa, that you need to justify why only elephant was included in the analysis. Particularly, this is important to justify as the variance in the tourism model is fairly low (39%).

2) Biodiversity is not the only attractor of tourists to protected areas. Activities can also be an important factor affecting visitation rates. This is the case of South Africa for example. I am not saying activities in the parks should be included in the analysis, but their importance should be at least discussed in the text.

3) I think the Methods are too complicated for readers without a background in Bayesian modelling and economic analysis. This makes the paper less appealing to the general readership of a journal such as Nature Communications. I suggest that the authors improve the explanation both in the main text and the Methods (see comments below). Additionally, I suggest that the authors add a flow chart so that the whole analysis can be visualized better.

4) I believe the example of Addo is misleading because of the artificial system (fences + culling + contraception) that has limited number of elephants in the park. The same limitation applies to other South African protected areas. Hence, this is a rather weak validation for your study. I'd remove it or be more straightforward about the artificial management system in Addo.

SPECIFIC COMMENTS

Page 5, line 112: can you be more specific about how you calculated the 5% starting from Moore J, Balmford A, Allnutt T, Burgess N. *Biol. Conserv.* 117, 343-350 (2004)?

page 7, line 142: how did you calculate the equivalent to \$565/km² in USD 2016?

page 12, lines 270-271: were these predictor variables included as dummy variables (0-1)?

Having more information on the predictor variables would be important too (even if referring to another study)

page 14, lines 310-314: can you provide more details here? I think someone with less experience

in projecting variables to the future will find it difficult to follow. Particularly, (i) how did you change elephant densities and (ii) did you keep all other predictor variables static? If so why? pages 15-16, lines 337-345: can you explain this better? Particularly, point 1 how the PYKE data were included in the regression model

Reviewer #3 (Remarks to the Author):

Nature Communications Article Review:

Title: Estimating the economic losses to tourism in Africa from the illegal killing of elephants

Authors: Robin Naidoo, Brendan Fisher, Andrea Manica, Andrew Balmford

Reviewer: Glenn K Bush, Assistant Scientist, Woods Hole Research center

Reviewer's summary:

This is an interesting and though provoking study. The article addresses a conservation management issue of critical importance today, the protection of the African elephant, in the context of a key policy tool (nature based tourism) to demonstrate the returns on investment from healthy populations of iconic species that contribute to the visitation experience. The paper aims to compare the economic contribution of stable elephant populations to nature based tourism revenues in order to justify public expenditure on measures to further improve specific elephant conservation initiative. It is a matter of critical importance, given the urgency surrounding the rapid decline of the species due to the recent resurgence of illegal elephant hunting. The paper provides novel quantitative evidence to help make well informed policy choices over decisions to spend scarce public finance on conservation, setting those choices in a context of managing protected areas in the broad and dynamic setting of poverty alleviation and economic growth. Two key conclusions are developed for conservation managers and policy makers in that 1) although in the main there are net positive benefits from biodiversity conservation it is not always the case making a finical justification of biodiversity conservation, secondary to moral or social imperatives and 2) the financial returns to elephant conservation are positive in savanna biomes, but not in forest biomes due to the different tourism context (and presumably consumer interest).

Review Comments:

General:

Overall the paper is well written and easy to understand, the sections are generally well developed and concise, but could do with some minor revision to add some further detail to assist in reader comprehension, including better "signposting" in the introduction about what is

presented and clearer articulation of the hypothesis.

The modelling procedure is relatively complex and principally drawn from the econometric domain. Its presentation in a natural science journal could benefit with a little more elaboration in the methods section to make it more appealing to a non-specialist economics audience. A section synthesis of the pertinent literature on bio-economic modelling would be beneficial as would a statement to how the model described contributes to that literature and conforms (or is an extension of) current knowledge, aside from the fact that it is a novel application.

Recommendation: This work is publishable subject to minor revisions to the presentation of results and clarification of narrative points.

Specific comments:

Introduction:

The introduction could more clearly specify objectives of the paper beyond estimating benefits and comparing the costs of combating their illegal killing" or set out a hypothesis. Given the compelling results and conclusions this could be along the lines of "....to demonstrate that elephant specific conservation efforts can have a net positive economic return".

There is no discussion of the cost/benefit analysis (CBA) framework adopted to motivate the analysis. Benefits are defined as the costs avoided (lost economic returns) from tourism that might occur if elephant numbers further decrease; quite a cognitively challenging precept, particularly for someone without a background in economic analysis. One might also intuitively think of costs as only those related to tangible direct impacts or activities such as compensation to local communities for in crop damages, anti-poaching or law enforcement efforts. It is perhaps a systematic appreciation of how economists think about values, versus everyone else that is one of the leading factors in why (as the authors highlight), "...the tangible benefits of biodiversity conservation are rarely understood"!

In addition the valuation approach is not established until the results section and could do with being introduced earlier. In the last paragraph of the introduction for the first time presents the concept of "benefits" ("Even if we entirely ignore other benefits...."). What benefits and how do these relate to costs?

The distinction between costs and benefits and their analysis gives essential context for the reader to understand the CBA in the results section. Initially the benefits (estimated as the costs avoided in terms of losses to the national/continental tourism revenues) are estimated as a gross value. Subsequently more "costs" related to those of enforcement are calculated then one "cost" is subtracted from the other "cost" to get a net positive value. I can imagine this is confusing to non-economist where it might be logically concluded that a negative subtracted from a negative should result in the summation of the total value of the negatives. The authors need to specify the relevant context for CBA in that in the policy hierarchy benefits to one group or level in society

can be reformulated as costs to another. In this case the local level management costs of increased enforcement are subtracted from the national benefits to derive the net value.

I do understand the need to be concise, but some mention of the TEV concept might help here to orientate the reader to relevant CBA literature and help with reader's conceptual comprehension of how costs and benefits are formulated and compared to each other e.g. Hanley, N. and Barbier, E (2009) Pricing Nature. Edward Elgar Publishing Ltd.

Results:

page 5: estimates for lost tourism benefits are presented using a central figure of circa \$15, million per annum. Why limit your estimates to the central figure? Given the error margin an estimate of the high and low as well as mid values could be made to estimate a range of gross returns to elephant conservation initiatives.

Page 8: regarding the subset of the main model where tourist visits are regressed against park area and country GDP. I am curious as to the causal theory of change between GDP and visitor numbers in these models. What assumptions are made about GDP change and visitor numbers? Is it that higher GDP nations attract more visitors, based on an assumed link between GDP level and state of tourism infrastructure and services? Is such an assumption realistic? Should GDP per capita be used or GDP growth rate? For that matter why only GDP, surely factors such as poor security and governance might be more relevant reasons affecting tourist choices to visit national parks? For example in Rwanda during the late 1990's and early 2000's whilst having one of the lowest GDP's per capita having recently emerged from the genocide and civil war, had a vibrant tourism industry due to having a tight grip on security and organization, not so for neighboring DRC. Wouldn't factors such as public expenditure on tourism and parks management be better predictors than GDP? I understand of course that any macro model can be improved upon and are necessarily built on best available data, but I think a discussion of the modelling framework and motivation for the choice of variables needed alongside some description of the constraints in data availability, this will also provide a useful point of discussion about further analysis.

Discussion:

Page 10 The two key conclusions might what to be reordered as the second conclusion is perhaps the most poignant (the financial returns to elephant conservation are positive in savanna biomes, but not in forest biomes due to the different tourism experience (and presumably consumer interest)).

What is currently presented as the first conclusion more naturally leads from above e.g. it is not always the case making a financial justification of biodiversity conservation, secondary to moral or social imperatives especially in the case of forest biomes

Page 11: the discussion on the value of illegal ivory echoes my earlier point from the results section on governance. Corruption of government officials plays a key part in the values of illegal ivory being realized by private individuals. In countries with weak governance and institutions e.g. fragile states it is difficult to separate public policy from private interest. Many of the weakest states of governance are found in the central African nations where elephant populations are found in forest biomes. Could that be the most important factor in the differences you see in your model. Could you re-run it with a governance factor for each country in the regression instead of or alongside GDP e.g. R. J. Smith, R. D. J. Muir, M. J. Walpole, A. Balmford & N. Leader-Williams (2003) Governance and the loss of biodiversity? *Nature* 426, 67-70.

Methods:

The methods section is quite well developed, logical and easy to follow, including sources and critical assessment of the sources, strengths and weaknesses of all data used in the model are adequately discussed. Some precision with regard to the modelling framework is required to make this study more easily replicable, as noted below.

The study develops a sophisticated bio-economic, multi stage, macro level model using secondary data collated from a range of primary sources. The authors describe the model as being estimated using multi step Bayesian parametric framework and from the description it appears that a Markov Chain Monte Carlo approach has been used. A short discussion of the relevance and benefits experienced by using a Bayesian approach over "traditional" regression approaches should be made with relevant references. This should lead to a formal econometric specification of the Bayesian regression model selected to avoid ambiguity about what exactly was done e.g. Green, W.H. (2012) *Econometric Analysis*. Prentice Hall, and provide sufficient evidence to motivate the choice of model in relation to data assumptions/constraints. If space in the main body of the paper is limited perhaps this could be included in supplemental information.

Page 13: In biological population estimation and modelling parametric uncertainty in the estimates of model parameters arises from a number of sources due to sampling variation, observer bias, and sampling error which can all manifest themselves in different ways depending on the estimation process used e.g. line transects, aerial counts, dung density counts etc. How will such factors affect the normality of the distributions across the many different studies you use, related to in motivating your choice of a parametric model? Can you comment on the robustness of your assumption? Did you assess each of the elephant population studies to look at this issue?

page 14: comment on how the required number of iterations in the MC chain was determined?

Page 15: as before, why 100,000 iterations of the MC

END.

Dear Dr. McKay,

Please find below a detailed response to the concerns and critiques of the three reviewers. We believe their comments have resulted in a manuscript that is substantially improved from the original submission. To summarize the main changes, we have:

- (1) Collected additional data on PA visits via a further literature review and contacts with experts in the field, and as a result have added an additional 22 PAs, including 13 PAs confirmed as having no visitors, thereby strengthening our inference for non-visited PAs (a concern raised by Reviewer #1).
- (2) Conducted extensive re-analyses, including an assessment of additional visit models that include the presence of lions (raised by Reviewer #2) and national-level governance (raised by Reviewers #1 and #3). We conclude that our core model + lion presence (but not governance) is the best model, and used this in all valuations in the paper.
- (3) Better described our analytical procedures, as raised by all 3 Reviewers, including expanded discussions of our cost-benefit framework, Bayesian statistical methods, and a new infographic that describes the workflow.
- (4) Included all input data into our visit model in a Supplementary Table (Reviewer #1).
- (5) Included as a separate file for review the results of standard multiple linear regression estimates by region (Reviewer #1).
- (6) Addressed other minor points as described in detail below.

Reviewer #1 (Remarks to the Author):

In summary: interesting study, worthwhile approach, definitely publishable, but as currently written, has some significant shortcomings which would need to be addressed through a fairly substantial revision.

Response – We thank reviewer #1 for their constructive and thoughtful comments on the manuscript. We have responded to the comments and revised the manuscript as detailed below.

The ms is framed as an economic calculation at regional scale, but its conclusions suffer from inadequate consideration of regional context and history. It is calculated using a private-actor micro-economic approach: short-term cash spent on anti-poaching, short-term cash received through tourism. So essentially it concludes that if (and only if) there are lots of wealthy tourists, relatively few elephants, and tourists differentially choose destinations with more elephants, then anti-poaching investments are profitable. That analysis, however, only applies at local scale. In fact, even where private conservation reserves do indeed have to make exactly that calculation for their anti-poaching investments, they take a much longer investment timeframe. At national scale, any form of tourism takes time to grow, to establish reputation and distribution networks and infrastructure. There is a wildlife tourism industry in East and South Africa because of anti-poaching investments in the past, and there will only be future wildlife tourism sectors in Central and West Africa if anti-poaching investments are made now. Of

course, the authors of this ms are very familiar with these issues, but the way the ms is presented, these key contextual issues are barely considered.

Response – We thank the reviewer for pointing out that our consideration of the above issues was deficient in the original submission. In response, we have now stated in the Discussion that tourism takes time to evolve in places and that responses to elephant decreases (or increases) will not happen immediately at any given site (p. 10 lines 221-224).

Neither the ms nor the Supplementary Information present the actual raw data. I think these should be provided. I would like to see the actual data, tabulated park by park, and including any secondary or estimated parameters calculated or interpolated from subsidiary models as in the SI.

Response – As requested we have now provided the input data to our Bayesian regression models, tabulated park by park, in Supplementary Table 2.

I also think that a single analysis including all parameters simultaneously runs substantial risk of analytical errors. Then I would like to see standard stepwise multiple linear regressions for each of the four regions, so we can judge the relative effects of different factors in different regions... Yes, linear regression models are unfashionable and Bayesian models are fashionable, but the difference between the underlying model philosophies are likely to have far less effect on the outputs than factors such as data deficiencies and transformations.

Response – As requested, we have now calculated separate multiple linear regressions for each of the four geographic regions that include the same variables as assessed in our hierarchical Bayesian model (see separate document attached). Note that because of the much smaller sample sizes when restricting analyses to individual regions (n=23-47) and consequent lack of power, many of the statistical relationships observed in the full model are not present in the regional models. For example, none of the predictor variables are statistically significant at the 0.05 level, nor does the model itself explain a significant percentage of the overall variance, for PAs in the West Africa region, due to the small sample size of 23. The same lack of significance is true for the Central region (n=47), which is not surprising given the much lower visitation rates in this region. Elephant density remains significant and positively associated with visits in the South (n=47), and with a marginally significantly positive interaction effect (though not a significant main effect) in the East (n=47). Models for the South and East also explained significant amounts of the total variance in tourism visits (19% and 39%, respectively). However, it should be noted that in addition to the sample size/power issues associated with running models separately by region, simple multiple linear regression is unable to account for the uncertainty in elephant population estimates that is present at each site, which is a key advantage of the Bayesian methodology (and which we now better describe on pages 15-16, lines 332-348). It's also important to note that while both reviewers 2 and 3 suggested that additional clarifications and explanations of the methods used were necessary, both supported the use of Bayesian methods in this application: "The authors have used similar statistical analyses and modelling techniques in previous papers and this is reflected in the fact that the Methods (for an expert) are sound and difficult to argue against" (Reviewer 2) and "The methods section is quite well developed, logical and easy to follow, including sources and

critical assessment of the sources, strengths and weaknesses of all data used in the model are adequately discussed... A short discussion of the relevance and benefits experienced by using a Bayesian approach over "traditional" regression approaches should be made with relevant references" (Reviewer 3). Therefore we believe that the best way forward is to retain the Bayesian analytical framework (though also modifying in response to comments from all three reviewers; see below) while better explaining the benefits thereof and the steps involved in the analysis, which all three reviewers stated would help improve the manuscript. We have done this on pages 15-16, lines 332-348, and have also included a Supplementary figure that is an infographic of our workflow (Supplementary Fig 4).

At present these regional differences are hidden in Table 2, which indicates that "cost to maintain population" is 120 times higher in Central than West Africa, which seems improbable...Currently, the conclusions for South and West Africa are surprising and very counter-intuitive. Therefore, they need to be supported very clearly and carefully by data and calculation which readers can follow, test and replicate.

Response – These costs are the costs required to maintain elephant populations that are >1000 individuals (it was large populations of roughly this size or greater for which the original Nigel Leader-Williams work estimated conservation costs). West Africa only has 2 such large populations that are experiencing relatively low levels of poaching, whereas Central Africa has 15 that have been very hard hit. This combination of factors is what is leading to the differences in cost estimates, since regional differences in habitat or in other factors were not considered by Leader-Williams. We have now stated this more clearly on p. 8-9 lines 179-183 when describing the ROI results.

For example, why is "year established" log-transformed (ST1)? Why not "years since establishment"? And how can the log of a year be 7.6? That means that the year is 10 (exp) 7.6 ?!!!

Response – Using years since establishment vs year established would not make a difference in the regression context since the variation across this variable would remain the same regardless of the scale used. We use the natural log (rather than base 10) in our transformations, hence $\exp(7.6) \sim 1998$ as year of PA establishment. However, we have now actually removed year of PA establishment in our revised modelling (see below).

How is "minutes to nearest city" measured? By road? Or by light aircraft, which is how the higher-spending tourists arrive? And why this particular measure, when data from tourism literature indicates that tourists will visit anywhere within 2 hours of a major airport, and visitation falls off heavily beyond that?

Response – Minutes to nearest city is measured via land or water, not air, and the reviewer's point is therefore well-taken. We have now clarified this at p. 14 lines 313-315. On the second point, we were not aware that the tourism literature suggests that tourists will visit anywhere within 2 hours of a major airport, with visitation falling off heavily beyond that. This second point seems counter to the first point, however, since light aircraft landing on remote airstrips are presumably not what the reviewer has in

mind in terms of a major airport. In any event, we still think our proxy for accessibility is relevant given that it should be correlated with air travel time, and indeed our empirical results show that this measure of accessibility is negatively associated with tourist visitation rates (i.e., the higher the travel time score, the lower the visits to a PA, all else equal), although support for this was not as high as with some of our other predictors (less than 90% CI).

How is attractiveness measured on a scale from 2.83 to 5.34?

Response - As described in Balmford et al. (2015), "... three experienced conservation scientists (Andrew Balmford, Andrea Manica, and Neil Burgess) independently scored (on a 1–5 scale) the likely attractiveness for nature-focused visitors of each of up to 14 biomes in each of eight terrestrial realms [23]. This score was based on the mammals and birds that might plausibly be seen by a visitor. There was reasonable agreement across experts in these scores (Spearman rank correlations on scores for N=65 biome-realm combinations: $r_s=0.85, 0.62, 0.51$; all $P < 0.001$). Each PA was then assigned the mean score for the biome-realm combination it occurred in (assessed by intersecting the WDPA, MHT and realm shapefiles). PAs which overlapped more than one biome received their mean score, but with +1 added to reflect the diversity of habitat types present."

Recognizing that all three reviewers commented they would like to see more detail on the Balmford et al. model that we modified, we have now added descriptions to the variables on p. 14-15 lines 311-319.

After safety and security, which are not considered at all, attractiveness is the dominant driver of tourist destination choice. But here it is treated equally with Country PPP, USD, log-transformed.

Response – We are not clear on what this comment means; all variables entered into the regression model are "treated equally". Our empirical results show that natural attractiveness did not have a significant impact on visitation rates to African PAs while Country PPP did have a positive effect on PA visits. Both of these results align with those for African PAs in the original Balmford et al. (2015) paper. Note that they did not consider safety and security in their original model, and as a result, neither did we in our original submission. However, in response to this comment and one of Reviewer #3, we have now used an expanded dataset (see below) to test three models of visit rates to PAs. In the first, we used the original Balmford et al. model and added the same three variables as in our original submission: date of PA establishment, elephant density, and forest/non-forest. In the second, we returned to the original Balmford et al model (removing PA establishment date), and added the elephant & forest variables as well as a variable for lion presence/absence. And in the third, we again returned to the original Balmford et al model, and added, in addition to the elephant & forest variables, both lion presence and a measure of country governance (the Ibrahim Index of African Governance). Calculating leave-one-out cross validation (loo) and the Widely-Applicable Information Criterion (WAIC) scores from the 'loo' package in R to compare models (Gelman et al. Statistics and Computing 2014; Vehtari et al. 2016 preprint at <http://arxiv.org/pdf/1507.04544.pdf>), we find that the new model containing lion presence/absence has a lower loo score and a lower WAIC score, and therefore (analogous to standard AIC comparisons with frequentist models) is a better fit than both the model from our original submission and the model containing both lion and governance. Consequently, we use

this new model for our valuation of tourism losses due to illegal elephant killing, as now described in an expanded modelling section in the Methods (pp.13-16, lines 285-363).

Why is elephant density NOT log-transformed, when the distribution appears to be highly skewed (minimum 0, mean 0.48, maximum 5.10)?

Response – We attempted a log transform of elephant density but it did not appreciably change the skewness of the data. In the Bayesian analysis context, a main impetus for transformation is to ensure that the scales of variables in a regression model are relatively similar, i.e., no huge differences in the magnitude of values that could result in difficulties in convergence of the Markov chains. Hence log-transforming larger variables such as year of establishment (in the 1950-2000 range) and country PPP (in the tens of thousands) is more important than those of variables with smaller values, like elephant density or natural attractiveness. Finally, we preferred to keep elephant density in its original units (individuals per km²) due to ease of interpretation of what density changes mean in terms of changes in park visits and associated monetary values.

Specifics.

Line 121. Economic losses driven by visitation rates not poaching rates. Standard problem with snapshot microeconomics. Eg, travel-cost valuations of parks, wilderness, forests etc appear higher for those where off-road vehicles, snowmobiles etc are permitted, because those are expensive, even though they create damage so conservation values are lower.

Response – What the reviewer is suggesting is certainly true. We interpret this comment to mean that the values that elephants provide are much greater than those associated with tourism, which we wholeheartedly agree with. However, we are not sure what to do to address this comment in the context here. We do note later in the paper (p. 12 lines 251-259) that funding mechanisms other than the capture of tourism values will be necessary to protect elephants in central Africa, and also that the use values that biodiversity provides are complements, rather than substitutes, to moral or aesthetic reasons for conservation (p 12 lines 259-263). We also now note that there are other economic benefits (and costs) that a wider-ranging, more comprehensive benefit-cost study could incorporate (p. 3 lines 55-65).

Line 153. "Savanna PA's of West Africa"? Which countries, which PA's? Very different from those of East and South.

Response – The PAs in west Africa are indeed different from those in east and southern Africa, but it's also true that any one PA will be different from others, even within the same region. Our intention in this paper was to determine whether any generalities regarding the economic value of elephants to tourism could be made, and as such we focus on this aspect of our results, rather than the inevitable

differences that exist among PAs that could be highlighted. Nevertheless, as mentioned above we now give brief reasoning as to why we see the ROIs that we do on p. 8-9, lines 179-183.

Line 210. What about African Parks?

Response – African Parks' website states that "African Parks is a non-profit conservation organisation that takes on direct responsibility for the rehabilitation and long-term management of protected areas in partnership with governments and local communities." Furthermore, "Where conditions allow, we stimulate the development of tourism enterprises as a means of increasing the economic and social impact that parks can have." Without a strict focus on elephants, and with tourism a core principle in the management of a number of their parks, we don't think it is as good an example of a funding mechanism that captures public concern and the existence value of elephants as the two we provide.

Line 240. What about EcoExist?

Response – The reviewer makes a good point that EcoExist does indeed do work on encouraging local community co-existence with elephants; we have added this information on p. 13 line 274-275.

Fig 2. Why has the regression line been forced through (0,0)? Why are the 8 points with $\log(\text{actual}) = 0$ included in the regression? Is the y-axis really $\log[(\text{actual visits})]$, or $\log[(\text{actual visits}) + 1]$? If the latter (which I suspect, since records are hardly likely to include a single visitor per annum), then the regression line should not be forced via (0,0), and the 8 zero-visitor parks should be excluded, which would yield a very different regression line, changing the entire analysis.

Response – The regression has not been forced through (0,0), as indicated by the regression equation given in the caption of the original submission that shows that the y-intercept is -0.171. The reviewer is correct to note that we should have labelled both Y and X axes as "...visits+1", since we did in fact add 1 to PAs that had zero visits in order to log-transform them. We have now done this in the revised Figure 2. Note that this figure simply shows the relationship between actual values and predicted values, i.e., it is a visual assessment of the goodness of fit or in-sample predictive strength of our Bayesian regression model. The fact that the line does in fact pass near the (0,0) origin and has a slope very close to 1 indicates that there is no systematic under- or over-prediction of the model against actual visits.

Whether removing the 9 zero-visitor parks would have yielded different coefficient values than those in Table 1 is an open question, but there seems no a priori reason to exclude from our analyses protected areas for which we have solid information that no visits occurred. Many protected areas will indeed have no tourism, and not including these protected areas in our models would bias our visitation estimates, and the impact of elephants on visits, upwards. However the reviewer does raise a good point regarding how few zero-estimate parks there were in our dataset. In response we have managed to expand our visitation database by 22 protected areas, including an additional 13 where tourist visits are zero. These have now been included in the revised model described in the text, and lends further confidence to our results being representative of the full spectrum of visitation levels of African parks in elephant range-state countries.

Fig 3. Why have industry sectors other than tourism been shown by individual country, but tourism only by region? Why are the ROI shown in increasing succession, rather than by sector or region?

Response - Our aim with this figure was to show a range of ROIs for investments that are typical in African elephant range state countries; for this we used the key references as stated in the manuscript, which had produced such values at the country level. Our intention was to illustrate that there is substantial variation in ROIs across countries and sectors, and to place our elephant conservation ROIs in context. We aggregate our ROI by region because there are simply not enough PAs within most countries to present meaningful values at the national level. We believe ordering our results from low-to-high best contextualizes where our elephant conservation values lie on a ROI spectrum, but could consider recrafting the figure along the lines that the reviewer suggests if it is a point that s/he feels very strongly about.

Reviewer #2:

I enjoyed reading this study that used Bayesian statistical modelling of tourist visits to protected areas to demonstrate how losing elephants from poaching would substantially decrease the benefits of elephant conservation to tourism in Africa's protected areas. The authors also assessed how these benefits compared to the costs of counteracting poaching. The authors have a strong publication record both in assessing the economic benefits derived from biodiversity and ecosystem services and in assessing factors affecting visitation rates in protected areas both in Africa and globally. The authors have used similar statistical analyses and modelling techniques in previous papers and this is reflected in the fact that the Methods (for an expert) are sound and difficult to argue against. However, I have a number of concerns that need to be addressed in a revised manuscript.

Response – We are grateful to the reviewer for their constructive and helpful comments on our paper, which we have attempted to address as described below.

1) The biggest assumption in the study is that elephant are the only attractor of tourists to Africa's protected areas. As the authors know very well, elephant is indeed one of the charismatic species tourists are after, but not the only one. Lion is at least as important as elephant for ecotourism and similarly declining throughout its range. So are other charismatic species (eg leopard). Hence, I wonder (i) whether your regression model should have included other charismatic species' density as a predictor variable and (ii) whether the estimate of loss is realistic considering you didn't include other charismatic species in the analysis? I think this is an important limitation of the study and needs to be explained better. I understand data are available mainly for elephant, partly because of the recent study by Wittemyer and colleagues that you cite, but I think, considering the literature on tourists' preferences in sub-Saharan Africa, that you need to justify why only elephant was included in the analysis. Particularly, this is important to justify as the variance in the tourism model is fairly low (39%).

Response – We agree that it's important to consider whether important, potentially confounding variables are missing from the model, such as charismatic species other than elephants. Ideally, density

estimates for other wildlife species that are important for tourism, such as lion and leopard, would have been available for Africa's PAs, as they are for elephants, but unfortunately this is not the case. To work around this, our original model of PA visitation included a subjective measure of the natural attractiveness of a PA's environment, based on the types of mammals and birds that might reasonably be seen by visitors (Balmford et al. PLoS Biology 2015). We again included this measure of mammal/bird attractiveness in the current model, which we have now explicitly stated on p. 14-15 lines 316-319.

Although density estimates for other charismatic megafauna are not comprehensively available, we have now expanded our model based on the reviewer's very good suggestion to include lion presence/absence at the 216 PAs that contain elephants, using data from recent rangewide lion papers (Bauer et al. PNAS 2015; Riggio et al. Biol. Cons. 2013) that summarize lion range occupancy across Africa. Given that Reviewers 1 and 3 also had suggestions on additional variables to test (specifically, country governance), we conducted the following re-analyses. First, we used the original Balmford et al. model and added the same three variables as in our original submission: date of PA establishment, elephant density, and forest/non-forest. Secondly, we returned to the original Balmford et al model (removing PA establishment date), and added the elephant & forest variables as well as a variable for lion presence/absence. And thirdly, we again returned to the original Balmford et al model, and added, in addition to the elephant & forest variables, both lion presence and a measure of country governance (the Ibrahim Index of African Governance). Calculating leave-one-out cross validation (loo) and the Widely-Applicable Information Criterion (WAIC) scores from the 'loo' package in R to compare models (Gelman et al. Statistics and Computing 2014; Vehtari et al. 2016 preprint at <http://arxiv.org/pdf/1507.04544.pdf>), we find that the new model containing lion presence/absence has a lower loo score and a lower WAIC score, and therefore (analogous to standard AIC comparisons with frequentist models) is a better fit than both the model from our original submission and the model containing both lion and governance. Consequently, we use this new model for our valuation of tourism losses due to illegal elephant killing, as now described in an expanded modelling section in the Methods (pp.13-16, lines 285-363). Note that the lion presence/absence variable has a 90% Bayesian credible interval above zero, indicating that all else equal, there is good evidence that PAs with lions do in fact experience higher visit levels than those without, as the reviewer had expected.

2) Biodiversity is not the only attractor of tourists to protected areas. Activities can also be an important factor affecting visitation rates. This is the case of South Africa for example. I am not saying activities in the parks should be included in the analysis, but their importance should be at least discussed in the text.

Response – This is another good point along the lines of whether the tourist visitation model includes all relevant confounding variables. We acknowledge that we have not included activities (we assume the reviewer is referring to excursions in/near a PA such as rafting, mountain biking, guided hikes, etc). As suggested, we have now explicitly mentioned that our model does not consider available activities at PAs as a potential predictor of PA visits (p. 15 lines 327-330). More generally, we have expanded our discussion on the Balmford et al. model in the Methods section on p. 14-15 lines 311-319.

3) I think the Methods are too complicated for readers without a background in Bayesian modelling and economic analysis. This makes the paper less appealing to the general readership of a journal such as

Nature Communications. I suggest that the authors improve the explanation both in the main text and the Methods (see comments below). Additionally, I suggest that the authors add a flow chart so that the whole analysis can be visualized better.

Response – As suggested, we have now attempted to better explain our statistical procedures and the concepts behind them in both the main text (pp. 3-4 lines 55-83) and in the Methods (pp.13-16, lines 285-363). See below for more specific responses to the helpful suggestions. Regarding the flow chart, we have now constructed a figure (Supplementary Fig 4) in the Supplementary Information that details our workflow as an infographic. We thank the reviewer for this excellent suggestion as we believe this figure will help readers better visualize the various steps in our analyses.

4) I believe the example of Addo is misleading because of the artificial system (fences + culling + contraception) that has limited number of elephants in the park. The same limitation applies to other South African protected areas. Hence, this is a rather weak validation for your study. I'd remove it or be more straightforward about the artificial management system in Addo.

Response – We have deliberated on this and ultimately believe we should keep the example, but with an explicit recognition of the limitations the reviewer has helpfully pointed out. Previous reviews of the manuscript have suggested a case study example that examines whether the patterns seen across space among the PAs in our dataset are also reflected over time within a PA. We agree that a better example of within-PA trends over time would provide stronger support for our across-PA results, but unfortunately the data at Addo seem to be the best available. We have noted the limitations mentioned above at p. 10 lines 210-213. We are also open to removing the example in our revised manuscript if the reviewer feels very strongly about its deletion, and if the Editor agrees.

SPECIFIC COMMENTS

Page 5, line 112: can you be more specific about how you calculated the 5% starting from Moore J, Balmford A, Allnutt T, Burgess N. Biol. Conserv. 117, 343-350 (2004)?

Response – Space constraints preclude us from going into much detail on this point in the manuscript, but we simply tabulated the ecoregion-level costs of effective biodiversity conservation presented in Moore et al. for all African ecoregions that contain elephants, which we now state explicitly on p. 6 lines 135-137.

page 7, line 142: how did you calculate the equivalent to \$565/km2 in USD 2016?

Response - We converted all figures to USD 2016 using the U.S. government standard deflator found at <http://data.bls.gov/cgi-bin/cpicalc.pl>. We have now included this link at p. 8 line 169.

page 12, lines 270-271: were these predictor variables included as dummy variables (0-1)? Having more information on the predictor variables would be important too (even if referring to another study)

Response – Forest/non-forest was included as a dummy variable, while the date (year) of PA establishment was included as a continuous variable (although this has now been removed due to the suggested additions of further variables to the core Balmford et al model). As mentioned above, we have now expanded the description of the model and the variables included on p. 14-15 lines 311-319.

page 14, lines 310-314: can you provide more details here? I think someone with less experience in projecting variables to the future will find it difficult to follow. Particularly, (i) how did you change elephant densities and (ii) did you keep all other predictor variables static? If so why?

Response – As we stated on lines 308 and 312 of the original submission, we did indeed keep all other predictor variables at their mean values, in order to isolate the impact of changes in predicted elephant densities on tourist visits. Elephant densities changed because to parameterize our model we used elephant population data from the same time period as our tourism data (mostly 1998-2007) wherever possible, while to predict densities at all 216 PAs we used the most recent population estimates (ca. 2009-2013). We have now noted this explicitly on p. 17 lines 367-372. Following this stage, the PIKE values, which estimate the fraction of an elephant's population that are being lost on an annual basis, were multiplied against the most recent elephant population estimate for each PA, resulting in a new, typically lowered population estimate as a result of poaching. We believe this section of the methodology should now be clearer from the text we have added as well as the suggested workflow in Supplementary Fig 4.

pages 15-16, lines 337-345: can you explain this better? Particularly, point 1 how the PYKE data were included in the regression model

Response – We have now attempted to better explain this point, by more clearly linking how PIKE estimates at each site result in an estimate of growth rate or change of elephants at each site (data from Wittemyer et al. PNAS 2014), on p. 18 lines 401-407. The Leader-Williams papers present elephant population changes as a function of anti-poaching cost/km²; we use this regression to back-calculate the expected anti-poaching costs at our set of PAs as a function of the current elephant population change under poaching pressure. We then estimate the shortfall in costs that would be needed to stabilize elephant populations (i.e. no increase or decline in population size) by subtracting the former figure from the Leader-Williams spend estimate of \$565km² in USD 2016 that is necessary to achieve no decline in elephant populations.

Reviewer #3

General:

Overall the paper is well written and easy to understand, the sections are generally well developed and concise, but could do with some minor revision to add some further detail to assist in reader comprehension, including better "signposting" in the introduction about what is presented and clearer articulation of the hypothesis.

The modelling procedure is relatively complex and principally drawn from the econometric domain. Its presentation in a natural science journal could benefit with a little more elaboration in the methods section to make it more appealing to a non-specialist economics audience. A section synthesis of the pertinent literature on bio-economic modelling would be beneficial as would a statement to how the model described contributes to that literature and conforms (or is an extension of) current knowledge, aside from the fact that it is a novel application.

Recommendation: This work is publishable subject to minor revisions to the presentation of results and clarification of narrative points.

Response – We thank Dr. Bush for his positive and constructive review of our paper, and believe his suggestions have made for a much stronger revision. See below for detailed responses.

Specific comments:

Introduction:

The introduction could more clearly specify objectives of the paper beyond estimating benefits and comparing the costs of combating their illegal killing" or set out a hypothesis. Given the compelling results and conclusions this could be along the lines of "...to demonstrate that elephant specific conservation efforts can have a net positive economic return".

Response – We agree with the reviewer on better "signposting", and have therefore modified the Abstract to more clearly and boldly call out the results of our analyses.

There is no discussion of the cost/benefit analysis (CBA) framework adopted to motivate the analysis. Benefits are defined as the costs avoided (lost economic returns) from tourism that might occur if elephant numbers further decrease; quite a cognitively challenging precept, particularly for someone without a background in economic analysis.

Response – We agree we could have done a better job of explaining the intuition behind the "benefits" that might be lost as a result of elephant declines from poaching, and we have now explicitly added this into the manuscript at p. 3-4 lines 67-73.

One might also intuitively think of costs as only those related to tangible direct impacts or activities such as compensation to local communities for in crop damages, anti-poaching or law enforcement efforts. It is perhaps a systematic appreciation of how economists think about values, versus everyone else that is one of the leading factors in why (as the authors highlight), "...the tangible benefits of biodiversity conservation are rarely understood"!

Response – The issue of crop damages is an important one to acknowledge, in that we did not address conflict costs or other (e.g., opportunity) costs that are relevant to local communities and others. We have now included this information as well as citations on p. 3 lines 60-65.

In addition the valuation approach is not established until the results section and could do with being introduced earlier.

Response – This is a good point, and we have now added text that introduces the general concept behind our valuation on p. 3 line 55-58 in the Introduction.

In the last paragraph of the introduction for the first time presents the concept of "benefits" ("Even if we entirely ignore other benefits..."). What benefits and how do these relate to costs?

Response – Benefits were actually mentioned in the second paragraph of the Introduction in the original manuscript). However, as described above and below, we now more clearly lay out our cost-benefit approach, and the limitations thereof with regard to a more comprehensive TEV approach, on p. 3 lines 55-65.

The distinction between costs and benefits and their analysis gives essential context for the reader to understand the CBA in the results section. Initially the benefits (estimated as the costs avoided in terms of losses to the national/continental tourism revenues) are estimated as a gross value. Subsequently more "costs" related to those of enforcement are calculated then one "cost" is subtracted from the other "cost" to get a net positive value. I can imagine this is confusing to non-economist where it might be logically concluded that a negative subtracted from a negative should result in the summation of the total value of the negatives. The authors need to specify the relevant context for CBA in that in the policy hierarchy benefits to one group or level in society can be reformulated as costs to another. In this case the local level management costs of increased enforcement are a subtracted from the national benefits to derive the net value. I do understand the need to be concise, but some mention of the TEV concept might help here to orientate the reader to relevant CBA literature and help with reader's conceptual comprehension of how costs and benefits are formulated and compared to each other e.g. Hanley, N. and Barbier, E (2009) Pricing Nature. Edward Elgar Publishing Ltd.

Response – We have now cited Hanley and Barbier in the context of TEV on p. 3 line 60, and have also addressed this point via the other changes described above.

Results:

page 5: estimates for lost tourism benefits are presented using a central figure of circa \$15, million per annum. Why limit your estimates to the central figure? Given the error margin an estimate of the high and low as well as mid values could be made to estimate a range of gross returns to elephant conservation initiatives.

Response – This is an excellent point and we would like to include further discussion in the manuscript on the range and end points of our estimates, but, given the other additions we have made, we feel that adding more here would raise the total word count while potentially distracting from the other main points we need to get across in the manuscript.

Page 8: regarding the subset of the main model where tourist visits are regressed against park area and country GDP. I am curious as to the causal theory of change between GDP and visitor numbers in these models. What assumptions are made about GDP change and visitor numbers? Is it that higher GDP nations attract more visitors, based on an assumed link between GDP level and state of tourism infrastructure and services? Is such an assumption realistic? Should GDP per capita be used or GDP growth rate?

Response – We include country GDP for the reasons that Dr. Bush suggests: we assume that PAs in richer countries that have greater levels of infrastructure and more comfortable and secure tourism environments overall are likely to have greater visitation rates, all else equal. In our prior work (Balmford et al. PLoS Biology 2015), we did in fact see a positive statistical relationship between visit rates and GDP (adjusted for purchasing power parity), as expressed by the positive coefficient on this variable in our model of tourism visits across African PAs.

For that matter why only GDP, surely factors such as poor security and governance might be more relevant reasons affecting tourist choices to visit national parks? For example in Rwanda during the late 1990's and early 2000's whilst having one of the lowest GDP's per capita having recently emerged from the genocide and civil war, had a vibrant tourism industry due to having a tight grip on security and organization, not so for neighboring DRC. Wouldn't factors such as public expenditure on tourism and parks management be better predictors than GDP? I understand of course that any macro model can be improved upon and are necessarily built on best available data, but I think a discussion of the modelling framework and motivation for the choice of variable is needed alongside some description of the constraints in data availability, this will also provide a useful point of discussion about further analysis.

Response – Dr. Bush raises a very valid point regarding our tourist visitation model and the set of predictor variables that we decided to include. Our approach in this paper was to retain the core variables from the model of Balmford et al., which had demonstrated predictive power in modeling tourist visits across African PAs, while adding several additional variables that were of particular relevance to the objectives of this study, most importantly that of elephant density. We recognize that we could have done a better job of justifying the inclusion of the original variables in the Balmford et al study; we have now done so on p. 14-15 lines 311-319. And we have included some additional discussion on the limitations of this type of modeling approach, including omitted variables and data inadequacies, on p. 15 lines 327-330.

Discussion:

Page 10 The two key conclusions might what to be reordered as the second conclusion is perhaps the most poignant (the financial returns to elephant conservation are positive in savanna biomes, but not in forest biomes due to the different tourism experience (and presumably consumer interest). What is currently presented as the first conclusion more naturally leads from above e.g. it is not always the case making a financial justification of biodiversity conservation, secondary to moral or social imperatives especially in the case of forest biomes

Response – We like this idea and as suggested, have reordered the two conclusions in the Discussion, on p. 11-12 lines 240-263.

Page 11: the discussion on the value of illegal ivory echoes my earlier point from the results section on governance. Corruption of government officials plays a key part in the values of illegal ivory being realized by private individuals. In countries with weak governance and institutions e.g. fragile states it is difficult to spearhead public policy from private interest. Many of the weakest states of governance are found in the central African nations where elephant populations are found in forest biomes. Could that be the most important factor in the differences you see in your model. Could you re-run it with a governance factor for each country in the regression instead of or alongside GDP e.g. R. J. Smith, R. D. J. Muir, M. J. Walpole, A. Balmford & N. Leader-Williams (2003) Governance and the loss of biodiversity? Nature 426, 67-70.

Response – In response to other suggestions on re-running the visitation model with lion as an additional predictor and with the paucity of zero-visit PAs in our data set, we performed the following set of re-analyses. First, we performed another literature review and reached out to more colleagues in Africa and from this, we were able to find visitor information on an additional 23 PAs, including an additional 13 parks where we could confirm no visitation occurs; this boosts the total number of zero-PAs to 22. Second, we used this expanded data set to test three models of visit rates to PAs. In the first, we used the original Balmford et al. model and added the same three variables as in our original submission: date of PA establishment, elephant density, and forest/non-forest. In the second, we returned to the original Balmford et al model (removing PA establishment date), and added the elephant & forest variables as well as a variable for lion presence/absence. And in the third, we again returned to the original Balmford et al model, and added, in addition to the elephant & forest variables, both lion presence and a measure of country governance (the Ibrahim Index of African Governance). Calculating leave-one-out cross validation (loo) and the Widely-Applicable Information Criterion (WAIC) from the 'loo' package in R to compare models (Gelman et al. Statistics and Computing 2014; Vehtari et al. 2016 preprint at <http://arxiv.org/pdf/1507.04544.pdf>), we find that the new model containing lion presence/absence has a lower loo score and a lower WAIC score, and therefore (analogous to standard AIC comparisons with frequentist models) is a better fit than both the model from our original submission and the model containing both lion and governance. Consequently, we use this new model for our valuation of tourism losses due to illegal elephant killing, as now described in an expanded modelling section in the Methods (pp.13-16, lines 285-363).

Methods:

The methods section is quite well developed, logical and easy to follow, including sources and critical assessment of the sources, strengths and weaknesses of all data used in the model are adequately discussed. Some precision with regard to the modelling framework is required to make this study more easily replicable, as noted below.

The study develops a sophisticated bio-economic, multi stage, macro level model using secondary data collated from a range of primary sources. The authors describe the model as being estimated using multi step Bayesian parametric framework and from the description it appears that a Markov Chain Monte Carlo approach has been used. A short discussion of the relevance and benefits experienced by using a Bayesian approach over "traditional" regression approaches should be made with relevant references.

Response – We have now explicitly identified the benefits of using a Bayesian regression approach over a traditional/frequentist multiple linear regression approach in the Methods section on p. 15-16 lines 332-348.

This should lead to a formal econometric specification of the Bayesian regression model selected to avoid ambiguity about what exactly was done e.g. Green, W.H. (2012) Econometric Analysis. Prentice Hall, and provide sufficient evidence to motivate the choice of model in relation to data assumptions/constraints. If space in the main body of the paper is limited perhaps this could be included in supplemental information.

Response – As suggested we have now formalized the model as per standard Bayesian notation (e.g. Gelman et al. 2013, McElreath 2015) in Supplementary Note 4.

Page 13: In biological population estimation and modelling parametric uncertainty in the estimates of model parameters arises from a number of sources due to sampling variation, observer bias, and sampling error which can all manifest themselves in different ways depending on the estimation process used e.g. line transects, aerial counts, dung density counts etc. How will such factors affect the normality of the distributions across the many different studies you use, related to in motivating your choice of a parametric model? Can you comment on the robustness of your assumption? Did you assess each of the elephant population studies to look at this issue?

Response – We did not assess in detail each of the 216 elephant population estimates, as this was beyond the scope of our study. The African Elephant Database contains the most up-to-date and comprehensive estimates of elephant populations across the continent and is the database used to make evidence-based policy decisions on issues related to the conservation of elephants in Africa. Each survey is vetted by country and/or regional experts from the IUCN SSC African Elephant Specialist Group before entry into the database, and a majority of population estimates also include error estimates as well as a central figure. Therefore we simply assumed the database represented the best available information on elephants and, as do other decision-makers regarding African elephant conservation issues, used these data to make economic inferences about the impact of poaching.

page 14: comment on how the required number of iterations in the MC chain was determined? Page 15: as before, why 100,000 iterations of the MC

Response – We now specify on p. 16 lines 355-356 that 100,000 iterations was determined as the number which led to stabilized resultant value estimations from preliminary trials.

East region (n=47)

Coefficients:

	Estimate	Std. Error	t value	Pr(> t)
(Intercept)	417.99887	202.13214	2.068	0.0457 *
log.area	0.07442	0.38286	0.194	0.8469
ele.dens	0.28615	0.40555	0.706	0.4849
forest	-1.56013	0.84483	-1.847	0.0728 .
ele.for.int[no.zeros\$region == "east"]	7.55669	3.97998	1.899	0.0654 .
nat.attrac	1.10963	0.34363	3.229	0.0026 **
log.pop	-0.19735	0.36343	-0.543	0.5904
log.ppp	2.72482	1.38493	1.967	0.0567 .
log.access.min	-0.13307	0.41882	-0.318	0.7525
log.year.est[no.zeros\$region == "east"]	-56.02497	27.12424	-2.065	0.0459 *

Signif. codes: 0 '***' 0.001 '**' 0.01 '*' 0.05 '.' 0.1 ' ' 1

Residual standard error: 1.513 on 37 degrees of freedom

Multiple R-squared: 0.5128, Adjusted R-squared: 0.3943

F-statistic: 4.327 on 9 and 37 DF, p-value: 0.0006696

Southern region:(n=47; elephant-forest interaction test not possible)

Coefficients:

	Estimate	Std. Error	t value	Pr(> t)
(Intercept)	110.18121	385.81919	0.286	0.77671
log.area	-0.17842	0.68270	-0.261	0.79520
ele.dens	1.51075	0.54830	2.755	0.00887 **
nat.attrac	-0.60479	0.66780	-0.906	0.37069
log.pop	-0.08715	0.41522	-0.210	0.83485
log.ppp	1.35008	0.86767	1.556	0.12779
log.access.min	-1.09954	1.14403	-0.961	0.34242
log.year.est[no.zeros\$region == "south"]	-13.45353	50.48773	-0.266	0.79128

Signif. codes: 0 '***' 0.001 '**' 0.01 '*' 0.05 '.' 0.1 ' ' 1

Residual standard error: 2.768 on 39 degrees of freedom

Multiple R-squared: 0.3131, Adjusted R-squared: 0.1898

F-statistic: 2.539 on 7 and 39 DF, p-value: 0.02968

West region: (n=23, elephant-forest interaction test not possible)

Coefficients:

	Estimate	Std. Error	t value	Pr(> t)
(Intercept)	-226.2218	630.1238	-0.359	0.725
log.area	-0.9557	0.7319	-1.306	0.213
ele.dens	2.0511	2.0702	0.991	0.339
forest	1.3468	2.4612	0.547	0.593
nat.attrac	1.2275	1.4002	0.877	0.395
log.pop	0.5441	0.4144	1.313	0.210
log.ppp	1.2481	2.1132	0.591	0.564
log.access.min	1.3062	2.0022	0.652	0.525
log.year.est[no.zeros\$region == "west"]	28.9143	84.1969	0.343	0.736

Residual standard error: 2.243 on 14 degrees of freedom

Multiple R-squared: 0.293, Adjusted R-squared: -0.111

F-statistic: 0.7253 on 8 and 14 DF, p-value: 0.6682

Central region: (n=47)

Coefficients:

	Estimate	Std. Error	t value	Pr(> t)
(Intercept)	9.839e+01	4.273e+02	0.230	0.819
log.area	-9.034e-01	1.065e+00	-0.849	0.402
ele.dens	-2.687e+00	1.796e+00	-1.496	0.143
forest	-9.378e-01	1.300e+00	-0.721	0.476

ele.for.int[no.zeros\$region == "central"]	5.301e+00	4.734e+00	1.120	0.270
close.area	-3.835e-05	3.855e-05	-0.995	0.326
nat.attrac	-5.089e-01	6.798e-01	-0.749	0.459
log.pop	-5.017e-01	6.240e-01	-0.804	0.427
log.ppp	1.368e-02	2.399e+00	0.006	0.995
log.access.min	-4.620e-01	1.280e+00	-0.361	0.720
log.year.est[no.zeros\$region == "central"]	-1.082e+01	5.652e+01	-0.191	0.849

Residual standard error: 2.887 on 36 degrees of freedom

Multiple R-squared: 0.2418, Adjusted R-squared: 0.03116

F-statistic: 1.148 on 10 and 36 DF, p-value: 0.3561

Reviewers' Comments:

Reviewer #1 (Remarks to the Author):

NCOMMS-16-14737A Second Review, Reviewer 1.

The authors have put substantial effort into addressing the reviewers' concerns.

I also acknowledge that Revs 2 and 3 have greater expertise than I in Bayesian statistics.

I recommend acceptance subject to the following minor revisions.

(a) Includes the multiple linear regressions (provided as an additional document for Rev 1 only) in the Supplementary materials.

(b) If editorial policies permit, include the methodological flowchart (Supp Fig 4) in the main text, since it is helpful to readers

(c) At a number of points in the text, insert the word "current" or "currently" to emphasise that tourism in forested biomes can change (and indeed, already is changing). These points are:

(i) Abstract, line 35, "although not currently in.."

(ii) Line 89, "although not currently so.."

(iii) Line 149, the aggregate current tourism..."

(iv) Line 251, "cannot currently be expected.."

(v) Line 253, "tourism levels are currently lower.."

(d) Line 221, ref 1, should also mention the very recent CITES census (publicised last week)

(e) Line 241, change "wise investment for tourism" to "wise investment with immediate and ongoing payback for tourism" The reason being that investing in elephant conservation is also wise in forested areas, where tourism is growing gradually. It is only the immediate payback which is different. Odzala-Kokoua NP in Congo-Brazzaville as an example.

Why are these changes important? Quite apart from academic concerns, if this article can be read as "don't bother about anti-poaching in forested ecosystems, it's not worth while" then it will be cited by development and hunting interests as an excuse to further decimate elephant populations in those regions.

Reviewer #2 (Remarks to the Author):

Thank you for addressing all of my concerns. I think the paper is now fit for publication.

Reviewer #3 (Remarks to the Author):

Second review:

Title: Estimating economic losses to tourism in Africa from the illegal killing of elephants.

Authors: Naidoo et al

Reviewer: Glenn Bush

The paper has been much revised as per the combined reviewers comments and is much clearer and stronger for it. The authors should be commended for turning round a profound set of review questions so quickly and thoroughly.

The changes to the paper as a result of my specific remarks have been well integrated and specific responses to my review points were clear and well justified.

One final observation: Table 1 shows various summary statistics for the model. Arithmetic means and standard deviations for dummy variables of “Forest” and “Lion” are tabulated. This does not make sense as Forest and Lion are discrete dummy variables (1,0), for park type or lion presence. I assume that the model results are automatically generated by the program. In addition the variables’ slope co-efficient are not reported.

I find that the paper is publishable as currently presented subject to the minor revision point above.

NCOMMS-16-14737A Second Review, Reviewer 1.

The authors have put substantial effort into addressing the reviewers' concerns. I also acknowledge that Revs 2 and 3 have greater expertise than I in Bayesian statistics.

We thank Reviewer 1 for their recognition of our attempts to address the many helpful comments of all three reviewers in round 1 of review.

I recommend acceptance subject to the following minor revisions.

(a) Includes the multiple linear regressions (provided as an additional document for Rev 1 only) in the Supplementary materials.

We believe that including both Bayesian and frequentist regression models would muddy the waters of the paper, both philosophically (since these are two quite different approaches) and empirically (since what the reviewer had asked us to produce is different to how we structured our analyses). However, since we have opted into the transparent peer review process, the linear regression results produced in response to Reviewer 1's original queries will appear online in association with the publication. We hope this is an acceptable compromise.

(b) If editorial policies permit, include the methodological flowchart (Supp Fig 4) in the main text, since it is helpful to readers

We have now included the original Supplementary Fig 4 as Fig 4 in the main text, as suggested.

(c) At a number of points in the text, insert the word "current" or "currently" to emphasise that tourism in forested biomes can change (and indeed, already is changing). These points are:

(i) Abstract, line 35, "although not currently in.."

(ii) Line 89, "although not currently so.."

(iii) Line 149, the aggregate current tourism..."

(iv) Line 251, "cannot currently be expected.."

(v) Line 253, "tourism levels are currently lower.."

(d) Line 221, ref 1, should also mention the very recent CITES census (publicised last week)

(e) Line 241, change "wise investment for tourism" to "wise investment with immediate and ongoing payback for tourism" The reason being that investing in elephant conservation is also wise in forested areas, where tourism is growing gradually. It is only the immediate payback which is different. Odzala-Kokoua NP in Congo-Brazzaville as an example. Why are these changes important? Quite apart from academic concerns, if this article can be read as "don't bother about anti-poaching in forested

ecosystems, it's not worth while" then it will be cited by development and hunting interests as an excuse to further decimate elephant populations in those regions.

We certainly agree with the Reviewer that we would like to avoid this perception and possible result, and therefore have inserted all of the suggested changes mentioned above.

Reviewer #2 (Remarks to the Author):

Thank you for addressing all of my concerns. I think the paper is now fit for publication.

We thank Reviewer 2 for their approval of all their concerns raised in round 1 of the review process. We believe that incorporating their comments have made for a stronger paper.

Reviewer #3 (Remarks to the Author):

Second review:

Title: Estimating economic losses to tourism in Africa from the illegal killing of elephnats.

Authors: Naidoo et al

Reviewer: Glenn Bush

The paper has been much revised as per the combined reviewers comments and is much clearer and stronger for it. The authors should be commended for turning round a profound set of review questions so quickly and thoroughly.

The changes to the paper as a result of my specific remarks have been well integrated and specific responses to my review points were clear and well justified.

We thank Dr, Bush for his many helpful comments in round 1, which we believe have resulted in a much stronger paper.

One final observation: Table 1 shows various summary statistics for the model. Arithmetic means and standard deviations for dummy variables of "Forest" and "Lion" are tabulated. This does not make sense as Forest and Lion are discrete dummy variables (1,0), for park type or lion presence. I assume that the model results are automatically generated by the program. In addition the variables' slope co-efficient are not reported.

Table 1 shows results of the posterior simulations for coefficients on all variables included in the Bayesian regression model. As such, these are not summary statistics of the variables themselves, but rather summaries of the coefficients on these models. We have now better clarified this in the table caption, which we believe led to this interpretation.

I find that the paper is publishable as currently presented subject to the minor revision point above.

We hope our explanation and clarification in the manuscript text has now clarified the above point.